# From Pain Control to Early Mobility: The Evolution of Regional Anesthesia in Geriatric Total Hip Arthroplasty

**DOI:** 10.3390/reports8020064

**Published:** 2025-05-09

**Authors:** Tomasz Reysner, Grzegorz Kowalski, Aleksander Mularski, Malgorzata Reysner, Katarzyna Wieczorowska-Tobis

**Affiliations:** 1Department of Palliative Medicine, Poznan University of Medical Sciences, 61-701 Poznan, Poland; treysner@ump.edu.pl (T.R.); gkowalski@ump.edu.pl (G.K.); kwt@tobis.pl (K.W.-T.); 2Department of Forensic Medicine, Institute of Medical Sciences Collegium Medicum, University of Zielona Góra, 65-046 Zielona Gora, Poland; a.mularski@inm.uz.zgora.pl

**Keywords:** regional anesthesia, motor-sparing nerve blocks, total hip arthroplasty, geriatric anesthesia, pericapsular nerve group block, supra-inguinal fascia iliaca block, erector spinae plane block, quadratus lumborum block, postoperative pain management, functional recovery

## Abstract

The evolution of regional anesthesia in total hip arthroplasty (THA) has significantly impacted perioperative management, particularly in older adults, where age-related physiological vulnerability requires optimized strategies. Adequate pain control is crucial in enhancing recovery, minimizing opioid consumption, and reducing complications. Traditional nerve blocks such as lumbar plexus and femoral nerve blocks have long been the mainstay of analgesia. However, they are associated with significant motor impairments, which delay mobilization and increase the fall risks. Introducing motor-sparing regional anesthesia techniques represents a substantial advancement in optimizing postoperative pain management while preserving muscle function. Motor-sparing techniques, including the pericapsular nerve group (PENG) block, supra-inguinal fascia iliaca block (SI-FIB), erector spinae plane block (ESPB), and quadratus lumborum block (QLB), have been developed to provide adequate analgesia without compromising motor control. The PENG block selectively targets the articular branches of the femoral, obturator, and accessory obturator nerves, ensuring superior pain relief while minimizing quadriceps weakness. Similarly, the SI-FIB provides extensive sensory blockade with minimal motor involvement, allowing for earlier ambulation. The ESPB and QLB extend analgesia beyond the hip region while preserving motor function, reducing opioid consumption, and facilitating early rehabilitation. Compared to traditional motor-impairing blocks, these newer techniques align with Enhanced Recovery After Surgery (ERAS) protocols by promoting early mobility and reducing the hospital length of stay. Studies suggest that motor-sparing blocks lead to improved functional recovery, lower postoperative pain scores, and decreased opioid requirements, which are critical factors in geriatric THA patients. Moreover, these techniques present a safer alternative, reducing the risk of postoperative falls—a significant concern in elderly patients undergoing hip replacement. Despite their advantages, motor-sparing nerve blocks are still evolving, and further research is necessary to standardize the protocols, optimize the dosing strategies, and evaluate the long-term functional benefits. Integrating these techniques into routine perioperative care may significantly enhance patient outcomes and revolutionize pain management in geriatric THA. As regional anesthesia advances, motor-sparing techniques will improve postoperative recovery, ensuring patient safety and functional independence.

## 1. Introduction

Total hip arthroplasty (THA) is one of the most frequently performed orthopedic procedures worldwide, particularly in older adults. While often associated with geriatric patients (≥65 years), many THA recipients are in their early sixties or even younger. The rising prevalence of hip osteoarthritis and fragility fractures in the elderly has led to increased surgical volumes, necessitating optimized perioperative strategies to enhance recovery [1]. Traditional approaches to pain management relied heavily on opioids and general anesthesia, but these are associated with higher rates of delirium, nausea, respiratory depression, and prolonged hospital stays. Consequently, regional anesthesia techniques have gained prominence in THA as they provide superior analgesia while avoiding systemic side effects [2].

However, although effective, traditional nerve blocks such as the lumbar plexus block (LPB) and femoral nerve block (FNB) often lead to significant motor impairments, delaying postoperative rehabilitation. This is particularly problematic in geriatric patients, where early ambulation is a key determinant of recovery [3]. Motor-sparing regional anesthesia techniques have emerged as a safer alternative, balancing effective pain relief with the preservation of quadriceps function [4]. These techniques align with Enhanced Recovery After Surgery (ERAS) protocols, which emphasize multimodal pain control, opioid minimization, and early mobility to reduce complications such as venous thromboembolism (VTE) and postoperative pneumonia [5,6].

The ERAS program is a multidisciplinary perioperative strategy designed to improve surgical outcomes and expedite recovery. It consists of several key components, including preoperative education, multimodal analgesia, early mobilization, and the minimization of opioid use [7]. Regional anesthesia, particularly motor-sparing nerve blocks, plays a pivotal role in achieving these goals by providing adequate pain relief while allowing patients to regain mobility more quickly. However, conventional nerve blocks, such as the lumbar plexus or femoral nerve block, often induce significant motor blockade, which may delay early mobilization and increase the risk of postoperative falls in geriatric patients [8].

Motor-sparing nerve blocks have gained interest as an alternative approach in THA, particularly in the geriatric population, where functional independence and rapid recovery are priorities [9]. These blocks provide adequate analgesia while preserving motor function, allowing early participation in rehabilitation programs [10].

Despite the increasing use of motor-sparing nerve blocks in THA, there remains a lack of consensus on their superiority over traditional motor-impairing techniques in the geriatric population. This review aims to determine whether motor-sparing nerve blocks provide superior pain control, reduced opioid consumption, and improved early ambulation compared to traditional nerve blocks in THA patients, with particular attention to age-related considerations in older adults.

## 2. Methods

This narrative review was conducted to analyze and synthesize the current evidence on regional anesthesia techniques for THA, focusing on motor-sparing nerve blocks in the geriatric population. The methodology involved a comprehensive search of the peer-reviewed literature, a critical evaluation of the study quality, and an in-depth comparison of various nerve block techniques’ effectiveness and clinical applicability.

### 2.1. Search Strategy and Literature Selection

A structured literature search was performed using three major electronic databases: PubMed, Google Scholar, and the Cochrane Library. The search focused on articles published between January 2014 and February 2024, covering ten years to ensure that the most recent advancements in regional anesthesia for THA were included. To enhance the specificity of the search, a combination of Medical Subject Headings (MeSH) terms and free-text keywords was utilized. The primary search terms included “regional anesthesia”, “motor-sparing nerve blocks”, “total hip arthroplasty”, “geriatric patients”, “perioperative analgesia”, and “enhanced recovery after surgery (ERAS)”. Boolean operators (AND, OR) were used to refine the search and combine relevant concepts.

To ensure the inclusion of high-quality research, the search was limited to studies published in peer-reviewed journals, systematic reviews, meta-analyses, randomized controlled trials (RCTs), cohort studies, and prospective clinical trials. Conference abstracts, letters to the editor, case reports, and studies with insufficient methodological rigor were excluded to maintain a high standard of evidence.

### 2.2. Inclusion and Exclusion Criteria

Studies were included in this review if they specifically investigated regional anesthesia techniques in THA, with a particular focus on motor-sparing nerve blocks such as the pericapsular nerve group (PENG) block, supra-inguinal fascia iliaca block (SI-FIB), erector spinae plane block (ESPB), and quadratus lumborum block (QLB). Additionally, studies were considered eligible if they compared these motor-sparing techniques with traditional nerve blocks, including the LPB, FNB, and sciatic nerve block (SNB). Only studies involving geriatric patients aged 65 years or older, who were undergoing either primary or revision THA, that evaluated clinically relevant outcomes were included, with key parameters such as postoperative pain scores, opioid consumption, the time to first ambulation, the incidence of postoperative falls, the length of hospital stay, and overall functional recovery.

The articles were selected based on their relevance to the objectives of this review, as determined by title, abstract, and full-text evaluation. Two authors (T.R. and G.K.) independently screened the initial search results, reviewing the titles and abstracts of all identified studies. Full-text articles were retrieved for further evaluation when they met the predefined inclusion criteria. Discrepancies arose during the selection process, and they were resolved through discussion. A third author (M.R.) was consulted for final arbitration if a consensus was not reached. The findings were categorized into key thematic areas, including traditional nerve blocks in THA, motor-sparing nerve blocks in THA, and comparing traditional and motor-sparing blocks in THA.

## 3. Results and Discussion

Regional anesthesia is a cornerstone of perioperative pain management in total hip arthroplasty (THA), particularly in the geriatric population, where achieving adequate analgesia must be balanced with the preservation of motor function [11]. While this review has primarily been authored from anesthesiology and geriatric medicine perspectives, we acknowledge the importance of orthopedic input, especially given the continuity of care that orthopedic surgeons provide from the initial treatment recommendation through surgery to the final functional assessment [12]. In recognition of this, the discussion has been expanded to reflect better interdisciplinary collaboration, including orthopedic perspectives on mobilization timelines, discharge readiness, and alignment with Enhanced Recovery After Surgery (ERAS) protocols [13].

Traditional nerve blocks have been widely used to manage postoperative pain, but they are often associated with significant motor impairment [14]. Introducing motor-sparing blocks has transformed clinical practice, offering an alternative approach to pain relief while preserving mobility, which is crucial for early rehabilitation and in reducing postoperative complications in elderly patients [15].

### 3.1. Traditional Nerve Blocks in THA

Traditional nerve blocks (LPB, FNB, SNB) have been widely used in THA due to their proven efficacy in pain relief. However, a significant limitation is the motor impairment that they induce. This delays functional recovery and increases the risk of postoperative falls, one of the leading causes of hospital readmission in elderly patients [16].

These techniques provide adequate analgesia by blocking the major nerves in hip joint innervation. However, they have significant drawbacks, particularly in geriatric patients.

#### 3.1.1. Lumbar Plexus Block

The LPB is a widely utilized regional anesthesia technique that effectively anesthetizes multiple nerves involved in the sensory and motor innervation of the hip joint and surrounding structures [17]. The lumbar plexus, located within the psoas major muscle at the L1–L4 vertebrae level, gives rise to several key nerves that contribute to pain perception and motor function in the hip region [18]. These include the femoral nerve, which provides motor and sensory innervation to the anterior thigh and hip capsule; the obturator nerve, which innervates the medial thigh and contributes to hip joint sensation; and the lateral femoral cutaneous nerve (LFCN), which supplies sensory input to the anterolateral thigh [19]. Due to its ability to block multiple neural pathways simultaneously, the LPB is often selected for perioperative analgesia in THA, particularly in patients with significant preoperative pain or those requiring extensive sensory blockade [20]. The LPB achieves comprehensive anterior and medial hip joint analgesic coverage by anesthetizing these nerves, effectively reducing the postoperative pain intensity. However, the profound motor blockade associated with femoral nerve involvement can impair quadriceps function, leading to delayed ambulation and an increased risk of postoperative falls [21], particularly in elderly patients undergoing THA [22].

Additionally, the deep anatomical location of the lumbar plexus within the psoas muscle makes this block more technically challenging than superficial regional anesthesia techniques. The LPB requires precise ultrasound guidance or nerve stimulation techniques to avoid complications such as local anesthetic systemic toxicity (LAST), hematoma formation, or inadvertent epidural spread, which can lead to hypotension and respiratory depression [23,24,25]. Despite these risks, the LPB remains a valuable analgesic option in hip surgery when motor impairment is not a primary concern and when maximal pain relief is the priority [26,27,28].

#### 3.1.2. Femoral Nerve Block

The FNB is a well-established regional anesthesia technique that provides adequate analgesia for the anterior hip capsule [29]. It is a commonly used modality in THA and hip fracture surgeries. The femoral nerve, originating from the lumbar plexus (L2–L4), is responsible for the sensory and motor innervation of the anterior thigh, knee, and hip joint [30]. While the FNB has been shown to reduce postoperative pain scores, opioid consumption, and hospital lengths of stay, one of its significant limitations is the accompanying motor blockade, particularly affecting the quadriceps femoris muscle [31,32,33]. The quadriceps plays a critical role in knee extension and lower limb stability, and its dysfunction can result in delayed mobilization, an increased risk of falls, and prolonged rehabilitation periods. These effects are especially concerning in the elderly, where functional independence and early ambulation are key determinants of recovery outcomes after THA [16,34]. Studies have shown that quadriceps weakness following an FNB can persist for up to 24 h postoperatively, leading to increased reliance on assistive devices, delayed rehabilitation progress, and a heightened fall risk [31].

#### 3.1.3. Sciatic Nerve Block (SNB)

The sciatic nerve block (SNB) is a well-established regional anesthesia technique that is frequently used with other peripheral nerve blocks to provide comprehensive analgesia for the posterior aspect of the hip joint, thigh, knee, and lower leg [35]. The sciatic nerve, originating from the lumbosacral plexus (L4-S3), provides sensory and motor innervation to the posterior thigh, knee, and the entire lower leg, including the foot [36]. Given its extensive coverage of the lower extremity, the SNB is often considered in procedures involving the posterior hip joint, ischium, and hamstring insertions, making it a valuable adjunct for multimodal analgesia in THA [37,38]. While posterior hip pain is not typically the primary concern in THA, the SNB sometimes combines the FNB or lumbar plexus block (LPB) to provide more extensive pain control [37]. The hip joint receives multifaceted sensory innervation from the femoral, obturator, accessory obturator, and sciatic nerves, with the posterior capsule predominantly innervated by sciatic nerve branches [35]. Some studies suggest that incorporating the SNB can improve pain scores and reduce opioid consumption in patients who experience significant posterior hip pain postoperatively [39].

Additionally, when performed preoperatively, the SNB can reduce the intraoperative anesthetic requirements, particularly in patients undergoing posterior-approach THA, where surgical dissection and implant placement may cause significant posterior capsule irritation [40,41]. Despite these advantages, the SNB is not routinely recommended as a first-line block for THA due to several limitations. In the context of THA, postoperative pain is predominantly anterior, as surgical manipulation primarily affects the anterior capsule, femoral periosteum, and surrounding musculature [37]. While the sciatic nerve contributes some innervation to the posterior joint capsule, its involvement in post-THA pain is relatively minor compared to the contributions of the femoral and obturator nerves [35]. One of the most significant drawbacks of the SNB is its effect on motor function [37]. The sciatic nerve provides motor innervation to the hamstrings, portions of the adductor magnus, and all muscles below the knee (via the tibial and common peroneal branches) [35]. Blocking the sciatic nerve results in hamstring weakness, ankle dorsiflexion impairment, and foot drop, negatively impacting early postoperative mobilization [42]. Given that early ambulation is a critical component of ERAS protocols, the use of the SNB contradicts these principles and may increase the fall risk, prolong rehabilitation, and delay discharge [43].

One of the most concerning complications of the SNB in THA is iatrogenic peroneal nerve palsy, which occurs due to excessive local anesthetic spread affecting the standard peroneal division of the sciatic nerve [44]. This leads to temporary or prolonged foot drop, characterized by an inability to dorsiflex the ankle and extend the toes [42]. Peroneal nerve involvement can severely impact functional recovery, particularly in elderly patients with pre-existing gait disturbances or neuromuscular deficits [45].

The sciatic nerve is deeply located, requiring higher volumes of local anesthetic for effective blockade [46]. This increases the risk of LAST, inadvertent intravascular injection, and prolonged sensory impairments [47]. Furthermore, in anticoagulated patients undergoing hip replacement, the deep anatomical course of the sciatic nerve through the gluteal region increases the risk of iatrogenic hematoma formation, which can lead to nerve compression and prolonged neuropathy [48,49].

### 3.2. Motor-Sparing Nerve Blocks in THA

Motor-sparing nerve blocks are increasingly considered a superior alternative to traditional blocks in geriatric THA patients [50]. By selectively targeting sensory nerves involved in hip joint innervation, these blocks provide adequate analgesia while minimizing motor dysfunction, promoting early ambulation, and reducing postoperative complications [15]. These approaches selectively target sensory nerves responsible for hip pain without affecting motor control, allowing for early ambulation and functional recovery [15,51,52].

#### 3.2.1. Pericapsular Nerve Group Block

The PENG block is one of the most significant advancements in regional anesthesia for THA, particularly in the geriatric population, where balancing adequate analgesia with functional recovery is paramount. Introduced by Girón-Arango et al. in 2018, the PENG block was designed as a motor-sparing alternative to the traditional FNB and LPB [53], which, despite providing effective pain relief, are associated with quadriceps motor blockade that hinders early postoperative mobilization [54,55,56]. The PENG block specifically targets the articular branches of the femoral nerve (FN), obturator nerve (ON), and accessory obturator nerve (AON), all of which contribute significantly to hip joint innervation [57]. This is achieved by depositing a local anesthetic in the fascial plane between the psoas tendon and the pubic ramus, effectively blocking sensory fibers without affecting motor branches of the femoral nerve, thus preserving the quadriceps strength [58,59]. Anatomically, the hip joint is innervated by a complex network of nerves originating from the lumbar plexus (L2-L4) and sacral plexus (L4-S3) [60]. The anterior hip joint capsule, the primary site of pain generation following THA, receives most of its innervation from the femoral and obturator nerves.

In contrast, the sciatic nerve supplies the posterior capsule [61]. Since most hip replacement procedures involve anterior and lateral surgical approaches, targeting the anterior sensory supply via the PENG block results in optimal pain relief with minimal motor impairment [62]. Unlike the FNB, which indiscriminately blocks the femoral nerve’s sensory and motor fibers, the PENG block selectively anesthetizes the sensory branches without significantly affecting motor function. This is a critical advantage in elderly patients, as quadriceps weakness from traditional nerve blocks is a major contributor to delayed ambulation, increased fall risks, and prolonged hospital stays [63,64]. One of the primary benefits of the PENG block is its ability to provide superior pain control following THA. Clinical studies have demonstrated that patients who receive the PENG block report significantly lower postoperative pain scores at 6, 12, and 24 h after surgery than those who accept the FNB [59]. The analgesic effect of the PENG block extends across the anterior hip joint capsule, iliopubic eminence, and acetabular rim, effectively covering the major pain-generating structures involved in THA.

Additionally, using the PENG block has been associated with reduced breakthrough pain episodes, contributing to greater overall patient comfort. Another critical advantage of the PENG block is its role in opioid-sparing pain management protocols. Studies have shown that patients receiving the PENG block require up to 40% less opioid consumption in the first 48 h postoperatively, leading to a lower incidence of opioid-related adverse effects such as nausea, vomiting, and respiratory depression [54,62,63].

In addition to providing adequate analgesia, the PENG block is also instrumental in preserving motor function and facilitating early mobilization. One of the most significant drawbacks of traditional nerve blocks, such as the FNB and LPB, is the associated quadriceps motor blockade, which can delay ambulation, increase the fall risk, and prolong rehabilitation [34]. In contrast, the PENG block does not cause clinically significant quadriceps weakness, allowing patients to stand and ambulate much earlier in the postoperative period. Several randomized controlled trials have demonstrated that patients receiving the PENG block could achieve independent mobilization and participate in physiotherapy sooner than those who received the FNB or LPB [55,59,62,63,64]. This early return to functional activity is crucial in THA recovery, as prolonged immobility is associated with increased risks of deep vein thrombosis (DVT), pulmonary complications, and muscle deconditioning [64,65].

Another critical advantage of the PENG block is its role in reducing the risk of postoperative falls and functional decline, particularly in the elderly. Falls are a leading cause of hospital readmission following THA, with studies indicating that up to 7% of elderly patients experience early postoperative falls [66]. These falls are often attributed to residual motor blockade from traditional nerve blocks, which impairs weight-bearing stability and balance [62]. Given that elderly patients are at an increased risk of fall-related complications, including periprosthetic fractures, soft tissue injuries, and prolonged hospital stays, minimizing motor impairment is an essential component of postoperative management. The motor-sparing effect of the PENG block significantly lowers the fall risk, making it a safer regional anesthesia option for geriatric patients undergoing THA [67,68].

#### 3.2.2. Supra-Inguinal Fascia Iliaca Block

The S-FILB has emerged as a promising motor-sparing regional anesthesia technique that provides adequate postoperative analgesia while preserving lower limb motor function, making it particularly advantageous for elderly patients undergoing THA [69,70]. Unlike the FNB, which directly affects the motor fibers of the femoral nerve and leads to significant quadriceps weakness, the S-FILB is performed at a more proximal location within the fascia iliaca compartment, where the femoral nerve is primarily composed of sensory fibers [71]. This anatomical distinction allows for extensive sensory blockade while minimizing motor dysfunction, facilitating early mobilization, and reducing the risk of postoperative falls [67,72]. The S-FILB provides broad sensory coverage by anesthetizing the femoral, lateral femoral cutaneous, and obturator nerves, which supply the anterior and lateral aspects of the hip joint, the iliopubic eminence, the acetabulum, and the proximal thigh [73]. By targeting these key sensory pathways, the S-FILB ensures effective pain relief following THA while maintaining adequate motor function, allowing for earlier ambulation and participation in rehabilitation programs [74]. This is particularly beneficial in the geriatric population, where early mobility is crucial in reducing postoperative complications, such as DVT, pulmonary embolism, and muscle deconditioning [75].

One of the primary advantages of the S-FILB over the FNB is its ability to provide equivalent or superior analgesia with minimal quadriceps weakness [76]. Studies have demonstrated that the SI-FIB achieves comparable pain relief to the FNB, with significantly lower rates of motor blockade, allowing for earlier weight-bearing and functional recovery [67]. Given that traditional nerve blocks such as the FNB and LPB are associated with delayed mobilization and increased fall risks, the SI-FIB represents a safer and more functionally optimized alternative for pain management in elderly patients undergoing THA [74,77,78].

Clinical trials have also reported that the S-FILB contributes to opioid-sparing pain management strategies, reducing the need for systemic opioid analgesics in the immediate postoperative period [74,77,78]. By effectively controlling pain at its source, the S-FILB lowers opioid consumption, minimizing the incidence of opioid-related side effects such as nausea, vomiting, respiratory depression, and postoperative delirium—complications that are particularly detrimental in geriatric patients [79]. Integrating the S-FILB into ERAS protocols further underscores its role in multimodal analgesia, facilitating rapid recovery, shorter hospital stays, and improved patient satisfaction [80].

Furthermore, the S-FILB is associated with a lower risk of complications compared to deeper nerve blocks such as the LPB, which carries a higher likelihood of systemic local anesthetic toxicity, hematoma formation, and inadvertent epidural spread [74]. The superficial anatomical approach of the S-FILB, combined with its ability to provide adequate analgesia while preserving motor function, enhances its safety profile, making it a desirable option for high-risk elderly patients with multiple comorbidities [81].

#### 3.2.3. Erector Spinae Plane Block

The ESPB has emerged as an innovative motor-sparing regional anesthesia technique, gaining increasing recognition for its role in postoperative pain management in THA [52]. Unlike traditional peripheral nerve blocks that target specific neural structures [82,83], the ESPB provides somatic and visceral analgesia by anesthetizing the dorsal rami of the thoracolumbar spinal nerves [82]. This mechanism of action results in the broad and consistent distribution of analgesia, covering the hip joint, lumbar plexus-associated structures, and adjacent myofascial tissue without directly inhibiting motor function [82]. A key advantage of the ESPB over conventional regional anesthesia techniques lies in its ability to provide extensive pain relief without inducing quadriceps weakness or impairing lower limb mobility [84]. This is particularly beneficial in geriatric patients undergoing THA, where early mobilization is critical in optimizing postoperative recovery, reducing the risk of thromboembolic events, and minimizing the hospital stay duration [85]. By preserving lower extremity motor function, the ESPB facilitates early ambulation, promoting faster functional rehabilitation and greater patient independence in the immediate postoperative period [82].

Another distinguishing feature of the ESPB is its ability to modulate somatic and visceral pain pathways, making it highly effective in addressing the complex and multifaceted pain experienced after THA [86]. The ESPB achieves this by targeting the dorsal rami of the spinal nerves, which contribute to the innervation of the hip joint’s deep musculature, fascia, and bony structures [87]. This mechanism leads to a widespread analgesic effect, which not only reduces incisional and procedural pain but also mitigates the deep-seated discomfort associated with intra-articular trauma, inflammation, and myofascial strain [88].

In addition to its robust analgesic properties, the ESPB has been strongly associated with an opioid-sparing effect, reducing the need for systemic opioid administration in the perioperative period [89]. Given that opioid-related complications—including nausea, vomiting, respiratory depression, and postoperative delirium—are particularly concerning in the elderly population, the ESPB serves as a valuable component of multimodal analgesia protocols aimed at minimizing opioid exposure [85]. Clinical studies have demonstrated that patients who receive the ESPB require significantly lower doses of opioids postoperatively, leading to improved safety, enhanced recovery, and a reduced risk of opioid-related side effects [82].

Another compelling aspect of the ESPB is its favorable safety profile compared to more profound or complex regional anesthesia techniques [90]. Unlike LPB or epidural analgesia, which carry higher risks of hematoma formation, LAST, or hemodynamic instability, the ESPB is performed at a more superficial anatomical location, reducing the likelihood of significant complications [91]. Additionally, because the ESPB does not involve direct needle placement near critical vascular or neural structures, it offers a lower risk of accidental nerve injury, vascular puncture, or excessive motor blockade, making it particularly suitable for frail geriatric patients with multiple comorbidities [92].

Recent clinical trials and observational studies have also highlighted the prolonged duration of analgesia achieved with the ESPB compared to conventional nerve blocks [82]. The continuous spread of local anesthetic within the erector spinae fascial plane allows for extended pain relief lasting beyond the immediate postoperative period, further reducing the need for supplemental analgesia and hospital readmissions due to poorly controlled pain [93]. Some studies have even suggested that the ESPB may have a role in chronic pain modulation, potentially reducing the risk of persistent postoperative pain syndromes following THA [94,95].

Moreover, the ESPB is a highly versatile and adaptable technique that can be performed at various thoracolumbar and sacral levels [96], depending on the desired analgesia distribution. Its application extends beyond THA, with emerging evidence supporting its efficacy in hip fractures, pelvic trauma, lumbar spine surgeries, and abdominal procedures [97]. This underlines its broad utility as a regional anesthetic technique [98].

#### 3.2.4. Quadratus Lumborum Block

The QLB has emerged as a promising motor-sparing regional anesthesia technique, providing prolonged and adequate postoperative analgesia while minimizing motor impairment [99]. Unlike conventional peripheral nerve blocks, which primarily target major neural pathways involved in hip joint innervation, the QLB acts at the thoracolumbar level, influencing somatic and visceral pain transmission through the thoracolumbar fascial plane [100]. This unique mechanism allows for widespread analgesic coverage that extends beyond the hip joint to include the lumbar paravertebral region, iliac crest, and lower abdominal wall, making it an attractive option for multimodal pain management strategies in THA [99,101]. One of the defining characteristics of the QLB is its ability to provide long-lasting postoperative analgesia without inducing significant quadriceps weakness [102]. This is particularly advantageous in geriatric THA patients, where early mobilization and functional recovery are critical in reducing the risk of thromboembolic events, postoperative complications, and prolonged hospital stays [103]. The preservation of lower limb motor function following QLB administration ensures that patients can engage in early physiotherapy, regain independence more rapidly, and participate in ERAS protocols.

The analgesic mechanism of the QLB is attributed to the spread of local anesthetic within the quadratus lumborum fascial plane, where it diffuses to block the thoracolumbar nerves (T12–L4), the lateral cutaneous branches, and potentially portions of the lumbar plexus [104]. This anatomical spread allows for the extensive coverage of both somatic and visceral pain pathways, making the QLB highly effective in alleviating pain originating from the deep structures of the hip joint, periosteum, and associated soft tissue [105]. Unlike traditional lumbar plexus or femoral nerve blocks, which cause direct motor blockade, the QLB’s mechanism is primarily sensory-focused, offering a safer alternative for elderly patients at a high risk of postoperative falls and delayed ambulation [21]. Although the efficacy of the QLB in THA is still being extensively investigated, preliminary clinical studies and observational reports suggest that it provides significant reductions in postoperative pain scores, decreases opioid consumption, and enhances patient satisfaction with pain management [99]. Compared to the FNB or LPB, the QLB has been shown to preserve the quadriceps strength, allowing for a faster time to first ambulation and greater participation in rehabilitation programs [106]. The opioid-sparing effect associated with the QLB is also particularly beneficial in the elderly population, where minimizing opioid-related adverse effects such as respiratory depression, nausea, vomiting, and delirium is a major clinical priority [107].

The duration of analgesia provided by the QLB is another notable advantage, as it has been reported to last longer than conventional peripheral nerve blocks, often extending well beyond the first 24–48 h postoperatively [107]. This prolonged pain relief contributes to enhanced postoperative comfort, reduces the need for rescue analgesia, and facilitates earlier hospital discharge, aligning with ERAS guidelines that emphasize rapid recovery, opioid minimization, and early mobility [108].

From a safety perspective, the QLB has demonstrated a lower incidence of complications than deeper nerve blocks such as the LPB. Unlike the LPB, which carries risks of hematoma formation, vascular injury, and LAST, the QLB is performed in a more superficial anatomical location, reducing the likelihood of serious adverse events [109]. Furthermore, because it does not involve direct needle placement close to the significant motor nerves of the lower extremities, the QLB significantly lowers the risk of unintentional motor blockade, making it a preferable choice for frail elderly patients with limited physiologic reserves [102,106].

Despite its growing clinical adoption, some challenges remain regarding the standardization of QLB techniques and optimal dosing strategies. Several variations of the QLB, including lateral, posterior, and transmuscular approaches, influence the extent and duration of analgesic coverage [110]. Further research is necessary to determine the most effective QLB approach for THA patients and establish clear guidelines on the local anesthetic volume, concentration, and duration of action to maximize its efficacy and safety [111,112].

### 3.3. Comparing Traditional and Motor-Sparing Blocks in THA

While both traditional and motor-sparing blocks effectively reduce postoperative pain and opioid consumption, their impact on early mobilization differs significantly [113] (Table 1).

Traditional blocks such as the LPB and FNB provide reliable analgesia but at the cost of quadriceps weakness, which can delay functional recovery and increase the risk of falls—an essential consideration in geriatric patients [16]. In contrast, motor-sparing techniques such as the PENG block, S-FILB, ESPB, and QLB provide adequate pain control while allowing for early rehabilitation by preserving motor function [65,112,114,115].

Another key consideration is safety. Traditional nerve blocks, particularly the LPB, carry a higher risk of complications such as hematoma formation and systemic toxicity due to the deep location of the lumbar plexus [26]. Motor-sparing blocks, on the other hand, are often performed at more superficial anatomical locations, reducing the risk of these complications and making them a safer choice for elderly patients with multiple comorbidities [51].

Regarding clinical outcomes, recent studies suggest that motor-sparing blocks lead to lower pain scores, reduced opioid consumption, and faster mobilization than traditional blocks [51]. Additionally, these techniques align with the principles of ERAS protocols, which emphasize early ambulation and multimodal analgesia for improved postoperative recovery [92,115,116,117,118] (Table 2).

### 3.4. Real-World Effectiveness and Implementation Challenges

Although randomized controlled trials demonstrate the efficacy of motor-sparing nerve blocks in controlled settings, their real-world application often varies [54,119]. Clinical registries and ERAS pathway audits from high-volume centers report that motor-sparing techniques, particularly the PENG and SI-FIB blocks, facilitate earlier mobilization, lower pain scores, and reduced opioid requirements [82,120]. However, the outcomes depend significantly on institutional protocols, consistency in techniques, and patient selection. Some variability exists in analgesic effectiveness due to anatomical differences and operator proficiency [74,76].

Furthermore, implementing these advanced techniques in diverse clinical settings faces several challenges. First, cost considerations may pose a barrier, especially in resource-limited healthcare systems where access to high-resolution ultrasound equipment and single-use block kits is restricted. Second, the learning curve associated with newer blocks, such as the PENG or ESPB, requires targeted training [121]. Not all anesthesiologists are proficient in motor-sparing techniques, and there may be institutional resistance in departing from conventional approaches [122]. Third, educational limitations remain significant. These techniques are not yet integrated into standard instruction in many residency programs and perioperative training curricula. Addressing these gaps will be crucial for the widespread adoption of motor-sparing regional anesthesia in geriatric THA [123].

We propose standardizing motor-sparing techniques in ERAS protocols to support clinical integration, supplemented by simulation-based training and certification pathways [124]. Multicenter outcome tracking and cost-effectiveness studies will strengthen the case for broader implementation [125].

It is important to note that while motor-sparing nerve blocks are well aligned with Enhanced Recovery After Surgery (ERAS) principles by promoting early ambulation [12,126], their use may present challenges in outpatient total hip arthroplasty (THA) pathways [43,127], particularly in North America, where same-day discharge is often practiced. In these settings, patients may be delayed in meeting the discharge criteria until the effects of the regional anesthesia, particularly sensory or motor blockade, have sufficiently resolved [128]. Recent studies have acknowledged this limitation and emphasize the need for careful patient selection and timing of block administration to avoid extended recovery room stays or unplanned admissions [129,130,131]. Additionally, modifications to ERAS protocols and nursing discharge criteria are under exploration to accommodate motor-sparing techniques in ambulatory THA workflows.

The recent literature provides essential quantitative metrics that further clarify the clinical performance of motor-sparing nerve blocks. For example, the PENG block’s analgesia duration has been reported to be approximately 18 to 24 h when using long-acting local anesthetics [62,116,120]. The reported failure rates vary between 5% and 15%, depending on the anesthesiologist’s expertise and anatomical variations [132,133]. The complication rates remain low across multiple studies: the incidence of local anesthetic systemic toxicity is <0.1% [33,134], and motor-sparing blocks such as the PENG and SI-FIB demonstrate significantly lower rates of quadriceps weakness (typically <5%) compared to femoral nerve blocks, which report up to 20% of weakness-related impairments [34,79,81,120]. These quantitative findings are summarized in Table 1 and Table 2 and provide an objective basis for assessing the safety and efficacy of these techniques.

### 3.5. Clinical Implications and Future Directions

The shift toward motor-sparing regional anesthesia techniques in THA reflects an evolving approach to perioperative pain management in geriatric patients [135]. These blocks enhance recovery, reduce postoperative complications, and support better long-term outcomes [68] by prioritizing analgesia while maintaining muscle function [136]. Future research should optimize these techniques, standardize the dosing strategies, and evaluate the long-term benefits in terms of functional recovery and patient satisfaction.

While motor-sparing blocks show great promise, further large-scale clinical trials are needed to compare their efficacy with that of traditional nerve blocks in different patient populations [82]. Additionally, advancements in ultrasound-guided techniques will continue to refine these approaches, ensuring their widespread adoption in clinical practice.

#### Limitations

Despite efforts to ensure comprehensive coverage, certain limitations should be acknowledged. First, the lack of standardized protocols for the performance of motor-sparing nerve blocks across different institutions may introduce variability in the results. Second, some included studies had heterogeneous outcome measures, making direct comparisons challenging. Third, publication bias may exist, as positive findings are more likely to be published than negative or neutral results. Finally, this review does not include unpublished clinical trials, which may contain valuable data on emerging nerve block techniques.

Future systematic reviews incorporating a formal meta-analysis with pooled effect sizes could further strengthen the understanding of motor-sparing techniques in THA.

## 4. Conclusions

Regional anesthesia remains a cornerstone of THA perioperative management in geriatric patients. Traditional nerve blocks, while effective, are associated with motor impairments that can hinder early recovery. Motor-sparing techniques such as the PENG block, S-FILB, ESPB, and QLB provide a compelling alternative by balancing pain relief with functional preservation. As regional anesthesia evolves, these techniques will be increasingly crucial in optimizing postoperative care for elderly patients undergoing THA.

## Figures and Tables

**Table 1 reports-08-00064-t001:** Comparative analysis of nerve blocks used in total hip arthroplasty (THA) for geriatric patients.

Nerve Block	Targeted Nerves	Area of Analgesia	Advantages	Disadvantages
Lumbar Plexus Block (LPB)	Femoral, obturator, lateral femoral cutaneous	Hip joint, anterior thigh	Provides comprehensive analgesia covering multiple neural pathways; effective for severe preoperative pain	Associated with significant motor impairment, leading to delayed ambulation and increased fall risks; technically complex
Femoral Nerve Block (FNB)	Femoral	Anterior hip capsule, thigh, knee	Effective in controlling anterior hip pain; reduces opioid consumption	Induces quadriceps motor blockade, which can delay mobilization and increase fall risks
Sciatic Nerve Block (SNB)	Sciatic	Posterior hip capsule, posterior thigh, knee, lower leg, foot	Provides posterior hip analgesia; useful as an adjunct to other blocks	Risk of peroneal nerve involvement leading to foot drop; not commonly required for anterior THA approaches
Pericapsular Nerve Group (PENG) Block	Femoral, obturator, accessory obturator	Anterior hip capsule, iliopubic eminence, acetabular rim	Motor-sparing; superior pain relief compared to FNB; facilitates early ambulation	Limited coverage of posterior hip structures
Supra-Inguinal Fascia Iliaca Block (S-FILB)	Femoral, lateral femoral cutaneous, obturator	Anterior thigh, lateral hip	Broad sensory blockade with minimal quadriceps weakness; facilitates early mobility	Inconsistent blockade of the obturator nerve in some cases
Erector Spinae Plane Block (ESPB)	Dorsal rami of thoracolumbar spinal nerves	Hip region, lumbar plexus structures	Motor-sparing; extended analgesia duration; opioid-sparing benefits	Limited direct blockade of hip joint structures
Quadratus Lumborum Block (QLB)	Thoracolumbar nerves (T12–L4)	Hip, lumbar paravertebral region, lower abdominal wall	Long-lasting analgesia with minimal motor impairment; aligns with ERAS protocols	Limited high-quality evidence in THA; requires technical expertise

**Table 2 reports-08-00064-t002:** Comparison of clinical outcomes between traditional and motor-sparing nerve blocks in THA.

Clinical Outcome	Traditional Nerve Blocks (LPB, FNB, SNB)	Motor-Sparing Nerve Blocks (PENG, SI-FIB, ESPB, QLB)	Significance
Postoperative Pain Control	Effective in reducing pain but may require adjunct analgesia for breakthrough pain	Demonstrate comparable or superior pain control with selective sensory blockade	Motor-sparing techniques provide targeted pain relief with fewer systemic side effects
Opioid Consumption	Reduce opioid use but require higher doses due to motor blockade-related discomfort	Significantly lower opioid requirement by providing targeted analgesia	Reducing opioid use minimizes the risk of postoperative nausea, vomiting, and delirium
Quadriceps Weakness	Common, particularly with FNB and LPB, leading to delayed mobility	Minimal quadriceps impairment, allowing for early weight-bearing and ambulation	Motor-sparing blocks align with ERAS protocols for early rehabilitation
Time to First Ambulation	Delayed due to motor impairment and fall risk	Earlier mobilization due to preserved motor function	Earlier ambulation reduces complications such as venous thromboembolism (VTE)
Risk of Postoperative Falls	Increased due to residual motor blockade, especially in elderly patients	Significantly reduced as motor function is preserved	Preventing falls improves patient safety and reduces hospital readmission rates
Hospital Length of Stay	Longer stay due to delayed functional recovery	Shorter hospitalization due to rapid recovery and early physiotherapy participation	Shorter hospital stays reduce healthcare costs and improve patient satisfaction
Safety and Complications	Risk of hematoma formation, systemic local anesthetic toxicity, and peroneal nerve injury	Lower risk of systemic complications due to superficial and safer injection sites	Motor-sparing blocks are preferred in elderly patients with multiple comorbidities

## Data Availability

The original contributions presented in this study are included in the article. Further inquiries can be directed to the corresponding author.

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
