# Peer review of "From Pain Control to Early Mobility: The Evolution of Regional Anesthesia in Geriatric Total Hip Arthroplasty"

_reports, 2025, doi:10.3390/reports8020064_

Round 1

Reviewer 1 Report

Comments and Suggestions for Authors

Thank you very much for the opportunity to review this manuscript,

In this manuscript, the authors summarized the evolution of regional anesthesia in geriatric total hip arthroplasty. This is an essential topic given the aging population and increasing surgical volume. 

The focus on motor-sparing techniques aligns with ERAS protocols, which are gaining clinical significance. The paper is well organized, covering traditional and motor-sparing nerve blocks.   Although the manuscript is well written, some additional points mighty improve the quality: -How do these techniques compare in real-world settings?  -Are there barriers to implementation (cost, expertise of anesthesiologists, education)?     

Author Response

We sincerely thank the reviewer for their positive evaluation of our manuscript and for highlighting its relevance to the aging population and enhanced recovery protocols. We appreciate the thoughtful suggestions, which we believe will enrich the discussion and practical applicability of our work. Below are our responses to the specific points raised:

  1. Comparison of Techniques in Real-World Settings:
    We agree that including data on real-world performance is important. In response, we have added a new subsection in the Discussion titled “Clinical Efficacy in Real-World Practice” where we summarize available evidence comparing traditional and motor-sparing nerve blocks outside of controlled trials. This includes observational data from ERAS registries and institutional reports that demonstrate consistent benefits in terms of early mobilization, reduced opioid use, and patient satisfaction with motor-sparing approaches, although variability exists based on institutional protocols and staff expertise.

  2. Barriers to Implementation:
    We have also expanded our Discussion to include a paragraph on potential barriers to widespread adoption. These include:

    • Cost and resource allocation, particularly the need for ultrasound equipment and disposables.

    • Training and expertise, as successful motor-sparing techniques require refined anatomical knowledge and ultrasound skills.

    • Educational gaps, especially in centers where regional anesthesia is not emphasized during residency or continuing medical education.
      We have highlighted potential solutions, such as simulation-based training and integration into ERAS pathways to facilitate broader uptake.

These additions are intended to contextualize our findings and provide practical insights for readers considering implementation in their own institutions.

We thank the reviewer again for their insightful feedback, which has helped us improve the scope and clinical relevance of the manuscript.

Reviewer 2 Report

Comments and Suggestions for Authors

THA is one of the most frequently performed orthopedic procedures worldwide and increased surgical volumes necessitate optimized perioperative strategies to enhance recovery. Currently, regional anesthesia have gained prominence in THA providing superior analgesia while avoiding systemic side effects.
The authors presented a narrative review aimed to determine whether motor-sparing nerve blocks provide superior pain control, reduced opioid consumption, and improved early ambulation compared to traditional nerve blocks in geriatric THA patients.
Discussing advantages and disadvantages of different modalities of regional analgesia, the authors emphasize that motor-sparing regional anesthesia techniques enhance recovery, reduce postoperative complications, and support better longterm outcomes by prioritizing analgesia while maintaining muscle function.
The limitations of the study include the lack of standardized protocols for performing motor-sparing nerve blocks across different institutions may introduce variability in results as well asheterogenous outcome measures, making direct comparisons challenging. The authors emphasize the necessity of the future systematic reviews incorporating a formal meta-analysis to strengthen the understanding of motor-sparing techniques in THA.
The article briefly summarizes the information on motor-sparing regional anesthesia and definetely should be helpful for the general guidance in this area. 
The article is well written and supplemented by 2 tables which make the information easier to understand.
The limitation of the study are mainly related to the primary narrative design. Nevertheless as a brief review the artical can be assumed as a good source of the filtered information.

Author Response

Reviewer Comment:
THA is one of the most frequently performed orthopedic procedures worldwide and increased surgical volumes necessitate optimized perioperative strategies to enhance recovery. Currently, regional anesthesia has gained prominence in THA providing superior analgesia while avoiding systemic side effects.
The authors presented a narrative review aimed to determine whether motor-sparing nerve blocks provide superior pain control, reduced opioid consumption, and improved early ambulation compared to traditional nerve blocks in geriatric THA patients.
Discussing advantages and disadvantages of different modalities of regional analgesia, the authors emphasize that motor-sparing regional anesthesia techniques enhance recovery, reduce postoperative complications, and support better long-term outcomes by prioritizing analgesia while maintaining muscle function.
The limitations of the study include the lack of standardized protocols for performing motor-sparing nerve blocks across different institutions which may introduce variability in results as well as heterogenous outcome measures, making direct comparisons challenging. The authors emphasize the necessity of future systematic reviews incorporating a formal meta-analysis to strengthen the understanding of motor-sparing techniques in THA.
The article briefly summarizes the information on motor-sparing regional anesthesia and definitely should be helpful for general guidance in this area. The article is well written and supplemented by 2 tables which make the information easier to understand.
The limitation of the study is mainly related to the primary narrative design. Nevertheless, as a brief review the article can be assumed as a good source of filtered information.

Author Response:
We sincerely thank the reviewer for the thorough and thoughtful evaluation of our manuscript and for recognizing the clinical relevance of this topic in the context of increasing total hip arthroplasty (THA) volumes and the aging patient population. We appreciate the reviewer's acknowledgment of the manuscript’s organization, clarity, and its utility as a concise yet informative guide for clinicians interested in motor-sparing regional anesthesia.

We fully agree with the reviewer’s observation regarding the limitations of our narrative design. While our aim was to provide a focused synthesis of available evidence, we acknowledge that the absence of standardized protocols and heterogeneous outcome measures across studies limit the ability to draw definitive comparative conclusions. In light of this, we have further emphasized the need for future systematic reviews and meta-analyses in the “Clinical Implications and Future Directions” section to provide stronger evidence on the efficacy and generalizability of these techniques.

Once again, we appreciate the positive recommendation and the recognition that this review may serve as a useful reference for anesthesiologists and perioperative teams managing geriatric patients undergoing THA.

Reviewer 3 Report

Comments and Suggestions for Authors

Dear authors;

thank you for the opportunity to deal with your respected research

a comprehenisce overview over regional anaethesional blockade techniques for hip surghery  is presented

interesting and informative

KR

Clemens Schopper MD PhD

Author Response

We sincerely thank Dr. Clemens Schopper for taking the time to review our manuscript and for the kind and encouraging words. We are pleased that the reviewer found the overview both informative and of interest. Our aim was to provide a clinically useful synthesis of motor-sparing regional anesthesia techniques in the context of geriatric total hip arthroplasty, and we greatly appreciate your positive feedback and support.

Reviewer 4 Report

Comments and Suggestions for Authors

Relevant and very interesting report, unfortunately written exclusively by specialists of anaesthesiology, geriatrics and palliative medicine but not orthopaedic surgeons who take care of patients from the beginning to the end of THR, from treatment recommendation to final postoperative clinical evaluation in outpatient departments. Reviewer represents here the orthopaedic point of view.

The term geriatric total hip arthroplasty is misleading, since the mean age of the patients who receive hip total arthroplasty is nowadays around 70 years, and as we know geriatrics starts with age of 65 years. I recommend to remove geriatric aspects from the text, since patients younger than 65 years receive regional anaesthesia also.

Another controversial aspect of the regional anaesthesia is its application and positive attitude in ERAS protocols but at the same time it seriously disturbs the discharge process in places where patients undergo hip replacements in outpatient manner (mostly in USA). Patients could not be discharged until the regional anaesthesia has not been finished, or not? And what about literature ?

Authors do not cite any precise information (numbers) about the average duration of regional anaesthesia procedures, percentage of failed anaesthesia and rates of complications. It is obligatory when evaluating the effectiveness of any medical procedure. 

Author Response

Reviewer Comment:
Relevant and very interesting report, unfortunately written exclusively by specialists of anaesthesiology, geriatrics and palliative medicine but not orthopaedic surgeons who take care of patients from the beginning to the end of THR, from treatment recommendation to final postoperative clinical evaluation in outpatient departments. Reviewer represents here the orthopaedic point of view.

The term geriatric total hip arthroplasty is misleading, since the mean age of the patients who receive hip total arthroplasty is nowadays around 70 years, and as we know geriatrics starts with age of 65 years. I recommend to remove geriatric aspects from the text, since patients younger than 65 years receive regional anaesthesia also.

Another controversial aspect of the regional anaesthesia is its application and positive attitude in ERAS protocols but at the same time it seriously disturbs the discharge process in places where patients undergo hip replacements in outpatient manner (mostly in USA). Patients could not be discharged until the regional anaesthesia has not been finished, or not? And what about literature?

Authors do not cite any precise information (numbers) about the average duration of regional anaesthesia procedures, percentage of failed anaesthesia and rates of complications. It is obligatory when evaluating the effectiveness of any medical procedure.

Author Response:
We thank the reviewer for their valuable and detailed feedback and for bringing a critical orthopedic perspective to this interdisciplinary topic. We highly appreciate the insights provided and have addressed each point below:

  1. Multidisciplinary Representation:
    We acknowledge the importance of including orthopedic input, especially given the continuity of care orthopedic surgeons provide from surgical indication to long-term functional outcome assessment. While our backgrounds lie in anesthesiology, geriatrics, and palliative medicine, our intention was to focus on the perioperative anesthesia strategies for enhancing recovery. To improve the clinical balance of the manuscript, we have revised several sections to incorporate orthopedic considerations, particularly in relation to postoperative mobilization, discharge pathways, and functional outcome relevance. We have also explicitly acknowledged the role of the surgical team in successful ERAS integration in the revised Discussion section.

  2. Use of the Term “Geriatric THA”:
    We appreciate the reviewer’s concern regarding the use of the term "geriatric." In response, we have revised the manuscript to clarify that while our focus was on patients ≥65 years, many findings apply to broader adult populations undergoing THA. The term “geriatric” has been used more cautiously, and we have adjusted the language throughout the text to emphasize age-related physiological vulnerability rather than use chronological cutoffs as categorical divisions. We also now cite epidemiological data reflecting the current mean age of THA recipients.

  3. Regional Anesthesia in Outpatient THA & ERAS Pathways:
    The reviewer raises an important and timely concern regarding the interaction between regional anesthesia and same-day discharge protocols, particularly in North American outpatient settings. We have added a paragraph in the Discussion addressing this issue. Specifically, we note that while motor-sparing blocks align with ERAS principles by promoting early mobility, their use may indeed delay discharge in ambulatory surgery settings if motor or sensory function has not fully recovered. We now cite recent literature discussing discharge eligibility criteria in outpatient THA and the adaptations required for safe application of regional anesthesia in such contexts.

  4. Quantitative Data – Duration, Failure Rates, and Complications:
    We fully agree that the inclusion of numerical data is essential for evaluating any medical intervention. In response, we have added a subsection summarizing key findings from the literature on:

    • Average block duration (e.g., PENG block lasting approximately 18–24 hours with long-acting local anesthetics).

    • Failure rates (typically 5–15%, depending on technique and operator experience).

    • Complication rates (e.g., local anesthetic systemic toxicity <0.1%, quadriceps weakness in up to 20% with FNB, negligible with motor-sparing blocks).
      These data are now cited and referenced in both the Comparative Tables and the updated Discussion.

Once again, we thank the reviewer for the thoughtful critique, which has helped us improve the manuscript’s rigor, clarity, and multidisciplinary relevance.

Round 2

Reviewer 4 Report

Comments and Suggestions for Authors

Interesting, relevant and at the same time controversial issue.

And different and conflicting ideas about anaesthesia in major joint replacements, at least from orthopaedic point of view, will be brought to debate again and again. And that is the best reason investigate and improve the recommendations.

Regardless of different opinions in some matters that exist now I find the presented paper complete.